# *LINC00958* promotes proliferation, migration, invasion, and epithelial-mesenchymal transition of oesophageal squamous cell carcinoma cells

**Biqi Wang**◉, **Duo Tang**◉, **Zijia Liu, Qian Wang, Shan Xue, Zijie Zhao, Dongdong Feng, Chao Sheng, Jintao Li, Zhixiang Zhou***

Beijing International Science and Technology Cooperation Base of Antivirus Drug, Faculty of environment and life, Beijing University of Technology, Beijing, China

◉ These authors contributed equally to this work.
* zhouzhixiang@bjut.edu.cn

**Data Availability Statement:** All relevant data are within the manuscript and its Supporting Information files.

## Abstract

Oesophageal cancer is one of the deadliest cancers in the world. Oesophageal squamous cell carcinoma (ESCC) is the most prevalent histological type of oesophageal cancer. Oesophageal cancer has a poor prognosis because of its invasiveness. Thus, it is especially important to seek effective treatment methods. Research indicates that long non-coding RNAs (lncRNAs) play a significant role in the occurrence and development of oesophageal cancer. The aim of this study was to describe the role of *LINC00958* in ESCC. Bioinformatics and real-time quantitative polymerase chain reaction (RT-qPCR) methods were utilized to predict and verify the expression of *LINC00958* in ESCC. Related functional experiments, including cell proliferation, migration and invasion, were performed. In addition, a western blot and a dual luciferase reporter gene experiment were used to study the detailed carcinogenic mechanism of *LINC00958*. The results indicated there was a high expression of *LINC00958* in ESCC, which promoted proliferation, migration, invasion and Epithelial–Mesenchymal Transition (EMT) of ESCC cells, and this effect may be via regulating miR-510-5p.

## Introduction

Oesophageal cancer (EC) is the seventh most common cancer in the world. It is estimated that there are 604,100 new diagnoses of EC every year [1]. Oesophageal squamous cell carcinoma (ESCC) is the main histological type of EC. The occurrence of EC may be related to many factors, including smoking, drinking, eating salty food, and viral infections [2,3]. EC is one of the deadliest cancers because of its invasiveness and low survival rate. Data show that the 5-year survival rate for EC is only 15–25% [4]. The incidence of EC increases every year and is highest in Asia and Africa [5]. According to research statistics, the number of global cases of EC is expected to increase to 808,508 in 2035 [6]. Therefore, it is particularly important to

**Funding:** This study was funded in the form of grants by the National Natural Science Foundation of China (Grant Nos. 42077399 and 21677006) awarded to Zhixiang Zhou.

**Competing interests:** The authors have declared that no competing interests exist.

understand the mechanisms of the occurrence and development of EC, and to seek effective therapeutic targets.

Long non-coding RNAs (lncRNAs) are a type of non-coding RNA that are greater than 200 nt in length; lncRNAs play a variety of important biological roles in cells [7]. For example, lncRNAs compete with microRNA (miRNA) by functioning as competitive endogenous RNA (ceRNA), thereby regulating the downstream target of miRNA [8]. Currently, evidence shows that an abnormal expression of lncRNA is closely related to the occurrence, metastasis and stage of tumours [9]. One study showed that lncRNA MALAT1 regulates the expression of HMGB1 by competing with miR-129-5p, which promotes the development of colon cancer [10]. In ESCC, lncRNA NEAT1 regulates the proliferation and invasion of ESCC through the miR-129/CTBP2 axis [11].

A recent study showed that *LINC00958* is involved in the carcinogenesis of bladder cancer [12]. The high expression of *LINC00958* has also been shown to promote the occurrence and development of certain cancers, including bladder cancer, oral cancer, pancreatic cancer, gastric cancer, cervical cancer and glioma [13–18]. However, the relevant mechanism of *LINC00958* in EC, especially ESCC, has not yet been determined.

In present study, we determined the upregulated expression of LINC00958 in ESCC cell lines, and functional studies showed that LINC00958 exerted enhanced actions on proliferation, migration, invasion and EMT of ESCC cells. In addition, we verified miR-510-5p, a tumour suppressor [19] to be a target miRNA of LINC00958, and suggested that the antitumor effect of miR-510-5p might be related to its target gene SPOCK1 [20]. Therefore, it is our expectation that the results of this study will provide new strategies for the treatment of EC.

## Materials and methods

### Cell culture and transfection

Three ESCC cell lines, EC109, EC9706 and KYSE180, and an immortal oesophageal epithelial cell line, Shantou Human Embryonic Oesophageal Epithelial (SHEE) cell line, were obtained from the Institute of Virology of the Chinese Academy of Preventive Medicine, Beijing, China. The cells were cultured in Gibco Dulbecco's Modified Eagle Medium (DMEM; Thermo Fisher Scientific Inc., Waltham, MA, USA) with 10% foetal bovine serum (FBS; Thermo Fisher Scientific Inc.) and HyClone penicillin-streptomycin (100 U/mL and 100 μg/mL, respectively) (Cytiva, USA), and placed in a 5% $CO_2$ cell culture incubator (Thermo Fisher Scientific Inc.) at a constant temperature of 37°C. The pcDNA3.1 plasmid that overexpressed the *LINC00958* gene was constructed and named pcDNA3.1-LINC00958. The empty pcDNA3.1 vector was regarded as the control. For the knockdown of *LINC00958*, siRNA-LINC00958-1 CCUUUG UUUCCAAAGGUUACC, siRNA-LINC00958-2 GCCUUAAAACUCACAUAGAGA and siRNA-LINC00958-3 GCGAAACUCCAUCUAAAAAAA (KeyGEN BioTECH, Nanjing, China) were designed and synthesised. The jetPRIME transfection reagent (PolyPlus transfection, Illkirch, France) was used according to the manufacturer's instructions for transfection. After 48 hours, the cells were collected.

### RNA extraction and real-time qPCR

Total RNA was extracted from the cells using Invitrogen TRIzol (Thermo Fisher Scientific Inc.) according to the manufacturer's instructions. The Ambion Protein and RNA Isolation System (PARIS™) kit (Thermo Fisher Scientific Inc.) was used to isolate nucleoplasmic RNA. Primers were found through the PrimerBank website (https://pga.mgh.harvard.edu/primerbank/) or designed using Oligo7 software (Table 1). The PrimeScript RT Reagent Kit with gDNA Eraser (Takara Bio Inc., Dalian, China) was used to synthesize 1 μg of total RNA

**Table 1. Primer sequences for real-time fluorescence-based quantitative-PCR.**

| Symbol | Oligonucleotide (5′ to 3′) | |
|--------|-----------|---|
| *GAPDH* | Forward | TGTTGCCATCAATGACCCCTTC |
| | Reverse | AGCATCGCCCCACTTGATTTTG |
| *U6* | Forward | CTCGCTTCGGCAGCACA |
| | Reverse | AACGCTTCACGAATTTGCGT |
| *LINC00958* | Forward | CACGTTTTATTTCTGACTGCT |
| | Reverse | AGTGGACTCATCTTTGCCT |

GAPDH, glyceraldehyde-3-phosphate dehydrogenase; U6, U6 small nuclear RNA; LINC00958, long intergenic non-protein coding RNA 958.

with a final volume of 20 μL to synthesise the cDNA. SYBR$^{®}$ Premix DimerEraser™ (Takara,) was used to detect gene expression using the ViiA 7 Real-Time PCR System (Thermo Fisher Scientific Inc.). GAPDH and U6 were used as endogenous controls for the cytoplasm and nucleus. The SHEE cell line was used as the experimental control. The 10-μL RT-qPCR reaction was repeated 4 times for each gene. The $2^{-\Delta\Delta Ct}$ method was used to calculate the relative gene expression levels.

## Wound healing assay

The wound healing assay was used to determine cell migration. A sterile 10-μL pipette tip was used to produce wounds in approximately 90% of the cells 24 hours after transfection. The cells were incubated at 37°C and 5% $CO_2$ for 24 hours. An inverted Axio Observer A1 fluorescence microscope (ZEISS, Jena, Germany) was used to assess the migration of cells into the wound at 0 hours and at 24 hours. The cell migration distance in μm was calculated using ImageJ software. The assay was repeated at least three times.

## Bioinformatics analysis

The bioinformatics-based ENCORI (The Encyclopedia of RNA Interactomes) website (http://starbase.sysu.edu.cn/) was used to predict the binding site of *LINC00958* and miR-510-5p [21]. The TargetScan website (http://www.targetscan.org) was used to predict the potential downstream mRNA targets of miR-510-5p. The GEPIA (Gene Expression Profiling Interactive Analysis) website (http://gepia.cancer-pku.cn/) was used to analysis data from TCGA (The Cancer Genome Atlas) [22].

## Dual luciferase reporter gene assay

The sequences of *LINC00958*-wt site and LINC00958-mut site were synthesised (Tsingke Biological Technology, Beijing, China) and constructed into the PmirGLO plasmid (Promega Corp., Madison, WI, USA), named PmirGLO-LINC00958 and PmirGLO-NC, and co-transfected with synthetic miR510-5p mimic (Tsingke Biological Technology, Beijing, China) into EC109 cells. After 48 hours, a Dual Luciferase Reporter Gene Detection kit (YuanPingHao Bio, Beijing, China) was used to detect the activity of luciferase according to the manufacturer's instructions.

## Clone formation assay

Forty-eight hours after transfection, the cells were seeded in a 6-well plate (Corning, New York, USA) at a density of 3,000 cells/well. After 14 days, the cells were washed twice with phosphate-buffered saline (PBS) (Thermo Fisher Scientific Inc., Waltham, MA, USA) and

fixed with 4% paraformaldehyde for 30 minutes. A 0.1% crystal violet staining solution was used to dye the cells for 10 minutes. The number of cell clones was then calculated.

## xCELLigence RTCA DP system

The xCELLigence Real-Time Cell Analyzer (RTCA) DP system (ACEA Biosciences, San Diego, CA, USA) was used to detect the effect that *LINC00958* had on the proliferation and invasion of the ESCC cell lines. Forty-eight hours after transfection, the cells were collected. To detect the invasion of the cells, 168 μL of DMEM with 10% FBS was added to the lower chamber of a CIM-Plate 16 (ACEA Biosciences), and 30 μL of DMEM without FBS was added to the upper chamber. The reference value was measured after the CIM-Plate was placed at 37˚C for 1 hour. The cells were seeded in a CIM-Plate 16 at a density of 30,000 cells/well. The medium in the upper chamber was supplemented to 100 μL, and performed for 24 hours; the invasion data was tested at regular time intervals. To determine the proliferation of the cells, 50 μL of DMEM with 10% FBS was added to an E-Plate 16 (ACEA Biosciences) to measure the reference value. The cells were seeded in the E-Plate 16 at a density of 2,000 cells/well. The medium in the upper chamber was supplemented to 200 μL, and performed for 72 hours; the proliferation data was tested at regular time intervals.

## Flow cytometry cell cycle analysis

Forty-eight hours after transfection, the cells were collected and washed twice with PBS and then fixed overnight in cold ethanol at 4˚C. Before the experiment, the cells were washed twice with PBS and incubated with RNase A (Thermo Fisher Scientific Inc., Waltham, MA, USA) for 30 minutes at 37˚C. The cells were then stained with propidium iodide A flow cytometer (FACScan system; BD Biosciences, San Jose, CA, USA) was used to analyse the stained cells.

## Western blot

Forty-eight hours after transfection, each group of cells was washed twice with ice-cold PBS; a RIPA lysis buffer (Cell Signaling Technology, Danvers, MA, USA) containing a protease inhibitor mixture (Roche, Basel, Switzerland) was used for processing. The sample was centrifuged at a rate of 12,000 rpm for 10 minutes at 4˚C, and the supernatant was extracted. A BCA protein assay kit (Solarbio Science & Technology Co., Ltd., Beijing, China) was used to measure the total protein concentration of the sample; the sample was diluted with PBS to ensure the concentration was consistent. A 5 × loading buffer (Applygen Technologies Inc., Beijing, China) was added, and the mixture was boiled in a water bath at 100˚C for 5 minutes. The protein sample was added to 10% sodium dodecyl sulphate polyacrylamide gel electrophoresis (SDS-PAGE) for processing. The semidry method was used to transfer the gel to the polyvinylidene fluoride (PVDF) membrane (Millipore Simga, Shanghai, China). A 5% skimmed milk powder diluted with Tris buffered saline with Tween 20 (TBST) (Applygen, Beijing, China) was used for sealing for 1 hour. The membrane was then incubated with a diluted primary antibody overnight. The primary antibody included: rabbit polyclonal antibody SPOCK1 (1:1000, BD-PN2270, Biodragon, Beijing, China); rabbit polyclonal antibody E-Cadherin (1:1000, #9782, Cell Signaling Technology); rabbit polyclonal antibody Vimentin (1:1000, #9782, Cell Signaling Technology); rabbit polyclonal antibody ZO-1 (1:1000, #9782, Cell Signaling Technology); rabbit polyclonal antibody Occludin (1:1000, #9782, Cell Signaling Technology); and rabbit polyclonal antibody GAPDH (1: 5000, 10494-1-AP, Proteintech). The membrane was washed 3 times with TBST for 10 minutes each time. The secondary KPL antibody was incubated with marked DyLight 800 (KPL, California, USA) for 1 hour. Finally, the

Odyssey infrared imaging system (LI-COR Biosciences, Lincoln, NE, USA) was used for the detection and quantification of the cells using ImageJ software.

## Statistical analysis

All data graphs were obtained using GraphPad Prism 7.04 software. All statistical analyses were performed using GraphPad Prism 7.04 and IBM SPSS Statistics software. The results were expressed as the mean ± standard deviation (SD). All data were analysed and compared using the $t$-test, ANOVA analysis or chi-square test. $P$ values < 0.05 were considered to be significant.

## Results

### *LINC00958* was upregulated in ESCC

To study the expression of *LINC00958* in ESCC, the GEPIA website was used to analyse the difference in gene expression between the tumour tissues and normal tissues. As expected, there was a high expression of *LINC00958* in 182 tumour tissues compared with 13 normal oesophageal tissues (Fig 1A). We used RT-qPCR to detect the expression of *LINC00958* in the ESCC cell lines, EC109, EC9706 and KYSE180. SHEE cells with a normal oesophageal cell phenotype was used as the control. We found a high expression of *LINC00958* in the ESCC cell lines (Fig 1B). The data indicated that *LINC00958* is upregulated in ESCC, which may be related to the development of EC.

### Downregulation of *LINC00958* inhibited cell proliferation and induced cell cycle arrest in the G1 phase

The knockdown and overexpression of *LINC00958* were used to verify the biological effects on ESCC cell lines. Three siRNAs were designed for the knockdown of *LINC00958*. We found

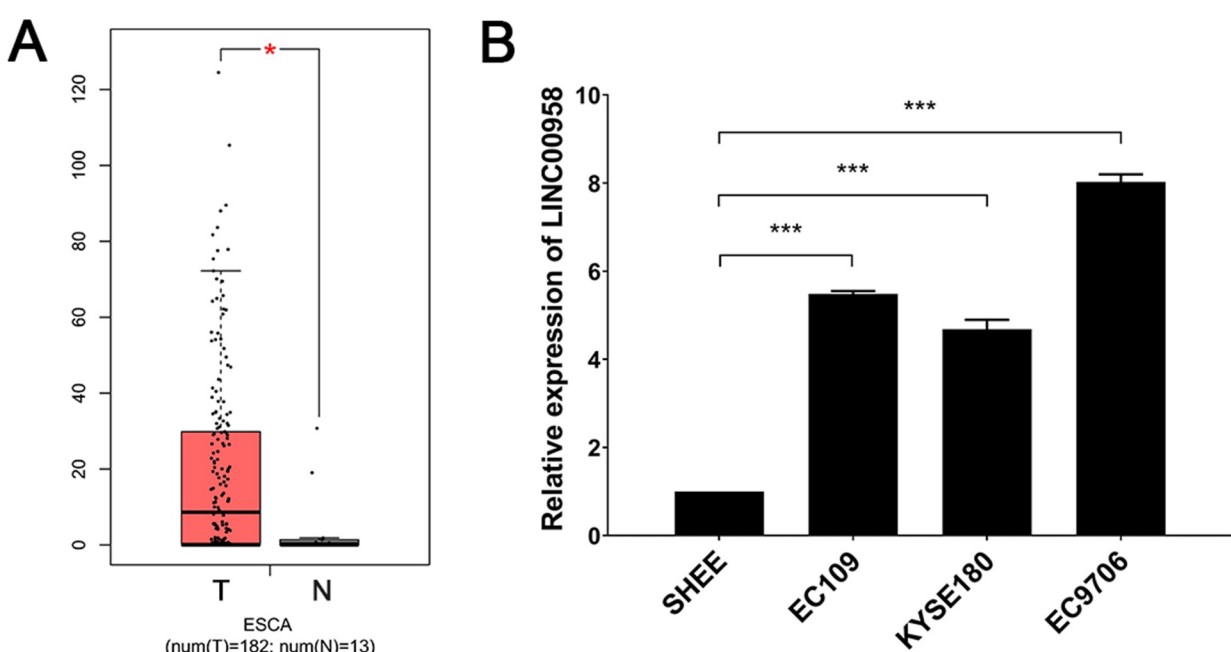

**Fig 1. Expression of *LINC00958* in ESCC cell lines and tissues.** (A) The mean expression of *LINC00958* in ESCA patients (GEPIA). The mRNA expression of pattern of LINC00958 in 182 ESCA tumour tissues (T) and 13 normal tissues (N) (* $P$ < 0.05). (B) The RT-qPCR detection of *LINC00958* in ESCC cells (*** $P$ < 0.001, n = 3).

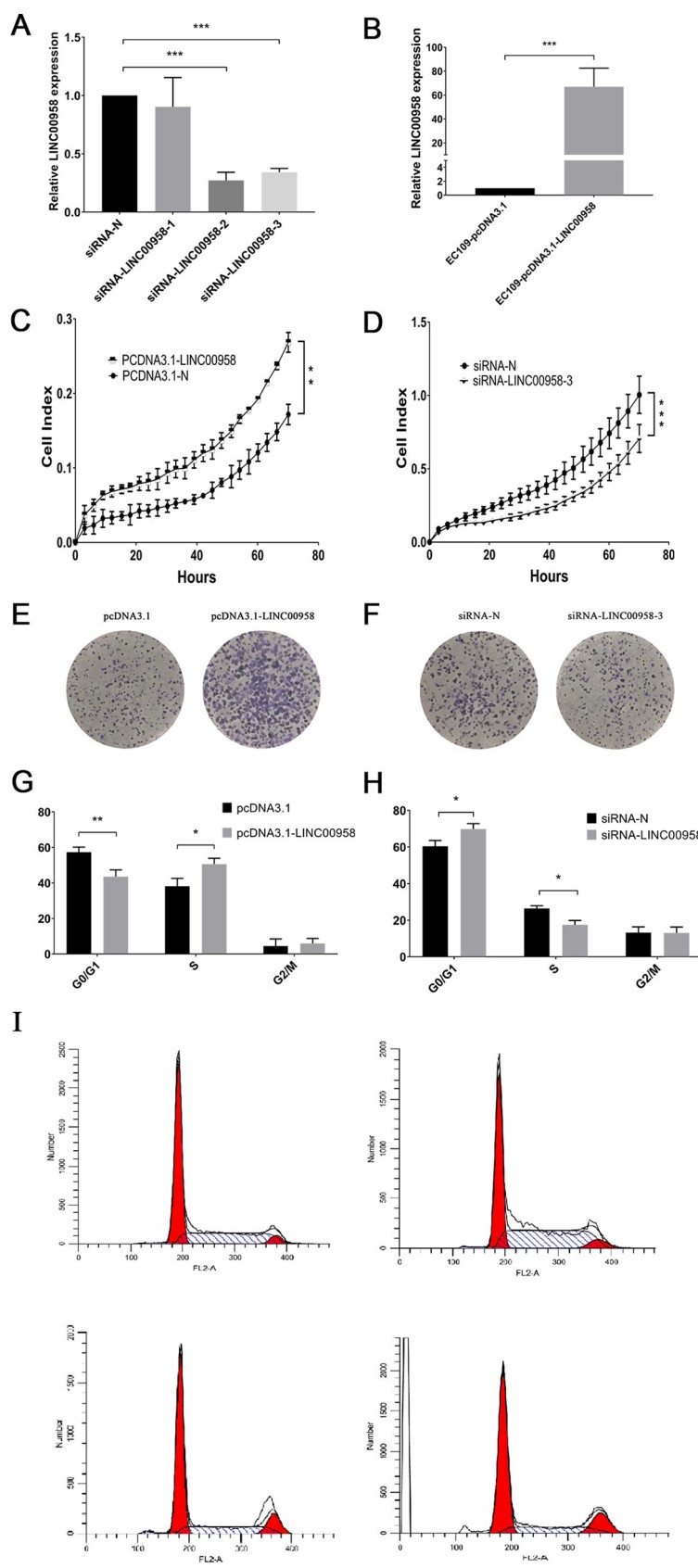

**Fig 2. Effects of *LINC00958* on the proliferation and cell cycle of ESCC.** (A) The expression of *LINC00958* after transfection of EC109 cells with siRNA. (B) The expression of *LINC00958* after transfection of EC109 cells with pcDNA3.1-LINC00958. (C and D) Real-Time Cell Analyzer DP system was used to detect the effect of *LINC00958* on the real-time proliferation of EC109 cells. (E and F) The effect of *LINC00958* on the clonal formation of EC109 cells. (G, H and I) Flow cytometry was used to detect the effect of *LINC00958* on the cell cycle of EC109 (* $P < 0.05$, ** $P < 0.01$ and *** $P < 0.001$, n = 3).

that siRNA-LINC00958-2 and siRNA-LINC00958-3 had significant knockdown effects (Fig 2A). After pcDNA3.1-LINC00958 was transfected into EC109 cells, the overexpression effect of *LINC00958* was significant (Fig 2B). The results showed that the overexpression of *LINC00958* promoted the proliferation of ESCC, which was detected using the xCELLigence RTCA DP system, while the knockdown of *LINC00958* inhibited the proliferation of ESCC (Fig 2C and 2D). Similar results were obtained in the clone formation assay (Fig 2E and 2F). In addition, the results of the cell cycle analysis showed that the number of cells in the G1 phase was significantly reduced due to the overexpression of *LINC00958*, while the number of cells in the S phase was significantly increased (Fig 2G and 2I). The knockdown of *LINC00958* caused a significant increase in the number of cells in the G1 phase and a significant decrease in the number of cells in the S phase (Fig 2H and 2I). These results indicated that the downregulation of *LINC00958* can induce cell cycle arrest in ESCC in the G1 phase.

## *LINC00958* regulated migration, invasion and EMT of ESCC

We verified the effect of *LINC00958* on migration and invasion of ESCC. The wound healing assay was used to determine the effect of *LINC00958* on the migration of the ESCC EC109 cells. The knockdown of *LINC00958* significantly inhibited the migration ability of ESCC, while the overexpression of *LINC00958* significantly promoted the migration ability (Fig 3A–3C). The xCELLigence RTCA DP system was used to measure the invasion of ESCC EC109 cells. After knockdown of *LINC00958*, the invasion ability of the EC109 cells was significantly reduced (Fig 3D). To determine whether *LINC00958* regulated the EMT of ESCC, we used a western blot to detect the expression of EMT-related proteins. The results showed that the downregulation of *LINC00958* could increase the expression of E-Cadherin, Occludin and ZO-1 and could decrease the expression of Vimentin (Fig 3E and 3F). The upregulation of *LINC00958* could reduce the expression of E-Cadherin, Occludin and ZO-1, and increase the expression of Vimentin (Fig 3G and 3H). These results indicated that *LINC00958* could regulate migration, invasion and EMT of ESCC.

## *LINC00958* regulated the expression of SPOCK1 by sponging miR-510-5p

To further study the function of *LINC00958*, we used the ENCORI website to predict the relationship between *LINC00958* and miR-510-5p (Fig 4A). A dual luciferase reporter gene experiment was used to verify that the miR-510-5p mimics significantly inhibited the luciferase activity of the PmirGLO-LINC00958 plasmid, but had no effect on the PmirGLO-NC plasmid (Fig 4B). Subsequently, we used the TargetScan website to predict the downstream mRNA of miR-510-5p, and we focused on SPOCK1. In addition, we predicted that there was a significant correlation between *LINC00958* and SPOCK1 in EC through the GEPIA website (Fig 4C). We also studied the influence of the expression of SPOCK1 in EC109 cells through the knockdown and overexpression of *LINC00958*. The western blot results showed that the expression of SPOCK1 significantly increased when *LINC00958* was overexpressed (Fig 4D and 4E), and significantly decreased when *LINC00958* was downregulated (Fig 4F and 4G). We synthesised miR-NC and miR-510-5p mimics and transfected them into EC109 cells. Compared with the

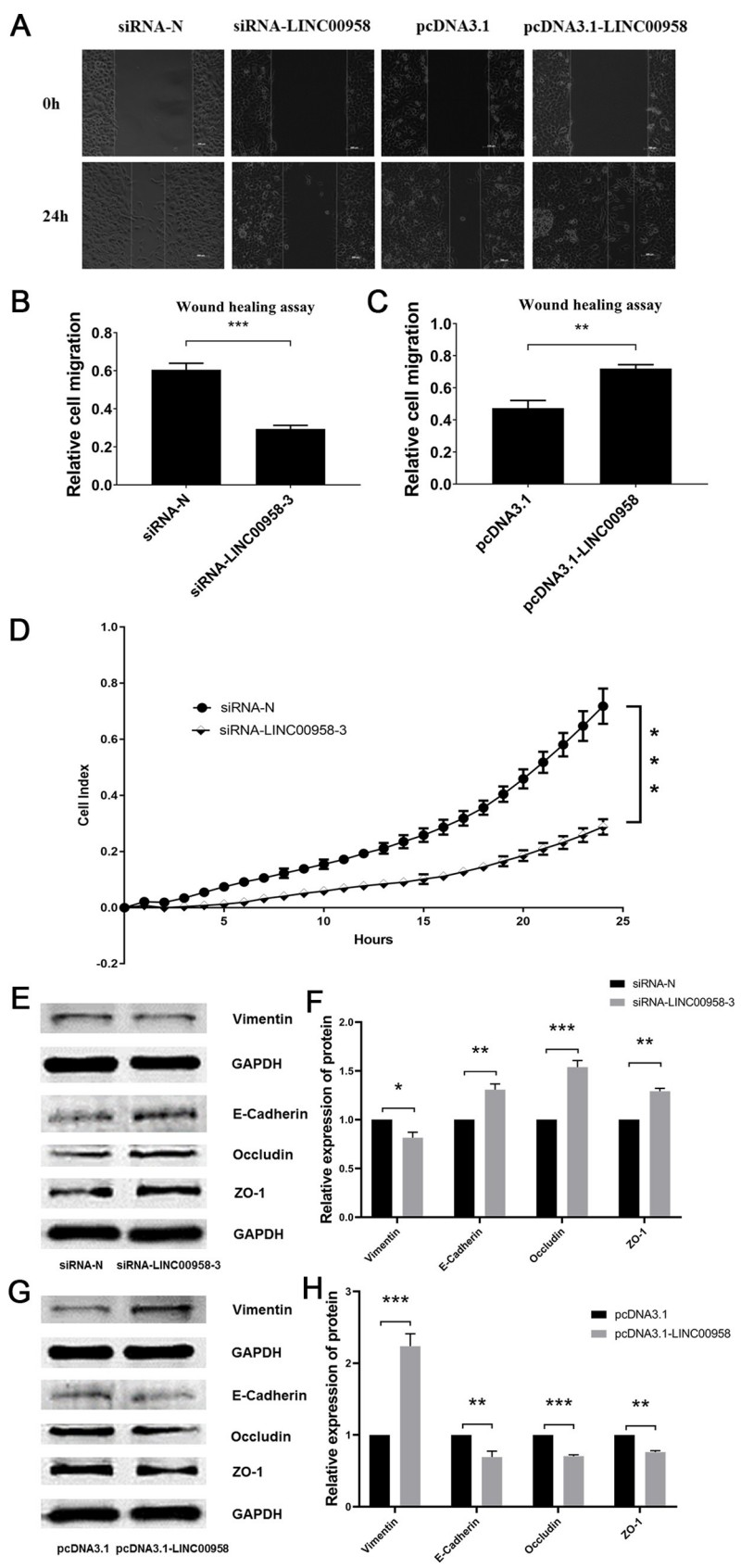

**Fig 3. Effects of *LINC00958* on migration, invasion and EMT of ESCC.** (A, B and C) The effect of *LINC00958* on EC109 cell migration was detected using a wound healing assay. (D) RTCA DP system was used to detect the effect of *LINC00958* on the real-time invasion of EC109 cells. (E, F, G and H) The effect of *LINC00958* on the expression of EMT-related proteins was detected using a western blot (* $P < 0.05$, ** $P < 0.01$ and *** $P < 0.001$, n = 3).

control group, miR-510-5p significantly inhibited the expression of SPOCK1 (Fig 4H and 4I). The results indicated that *LINC00958* could regulate the expression of SPOCK1 by competing with miR-510-5p in ESCC.

## Discussion

EC is one of the deadliest and most aggressive cancers. Currently, the main treatment is an oesophagectomy. However, at the late stage, EC cannot be cured by surgical resection due to its high invasiveness. Therefore, multimodality treatment methods are particularly important [23].

LncRNAs play an important biological role in the occurrence and development of cancer. They mainly compete with miRNA in the form of ceRNA [9,24]. Experiments have proven that lncRNAs also play an important role in EC. For example, lncRNA SNHG6 regulates the expression of HIF-1α by competing with miR-186-5p and, thus, promotes the proliferation, migration and invasion of EC cells [25]. *LINC00152* increases the expression of FYN by competing with miR-153-3p to promote the proliferation of EC cells [26].

In recent years, the function of *LINC00958* has been widely studied. For example, Chen et al. reported that *LINC00958* upregulates NUAK1 through sponging miR-625 to promote the malignant development of nasopharyngeal carcinoma [27]. Luo et al. found that *LINC00958* promotes the occurrence of non-small cell lung cancer by activating the JNK/c-JUN signalling pathway [28]. Zuo et al. confirmed that a high expression of *LINC00958* in liver cancer upregulates the expression of hepatic cancer-derived growth factor (HDGF) by competing with miR-3619-5p, thereby promoting the progression and adipogenesis of hepatocellular carcinoma [29]. Thus, *LINC00958* can be considered to be a therapeutic target for the systemic treatment of hepatocellular carcinoma using siRNA.

Although the general function of *LINC00958* in cancer has become clear, the specific role of *LINC00958* in EC has not yet been determined. In our current work, we found that *LINC00958* showed a high expression in EC tumour tissues through the TCGA analysis. We also detected an upregulated expression of LINC00958 in ESCC cells, and verified that *LINC00958* promoted proliferation, migration, invasion and EMT of ESCC.

To verify the function of *LINC00958* in ESCC, we predicted that there was a targeting relationship between *LINC00958* and miR-510-5p using the ENCORI website (S1 Fig). Studies have reported that miR-510-5p acts as a tumour suppressor in renal cell carcinoma and, thus, inhibits cell proliferation and migration, and promotes cell apoptosis [30,31]. It can be speculated that miR-510-5p may act as a tumour suppressor in ESCC and play a similar role to that in renal cell carcinoma.

SPOCK1 is a member of the SPARC family and plays an important role in cell proliferation, adhesion and migration [32,33]. Through the TargetScan website, we predicted that there was a targeting relationship between miR-510-5p and SPOCK1. Our experimental results showed that miR-510-5p is the upstream miRNA of SPOCK1. Currently, a number of studies have shown that SPOCK1 acts as an oncogene in various cancers. SPOCK1 promotes the proliferation and migration of cancer cells, regulates the cell cycle and inhibits apoptosis through the PI3K/Akt pathway [34–36]. The upregulation of SPOCK1 is also related to the advanced T stage or Gleason score of some cancers [37]. Research has also been done on SPOCK1-related

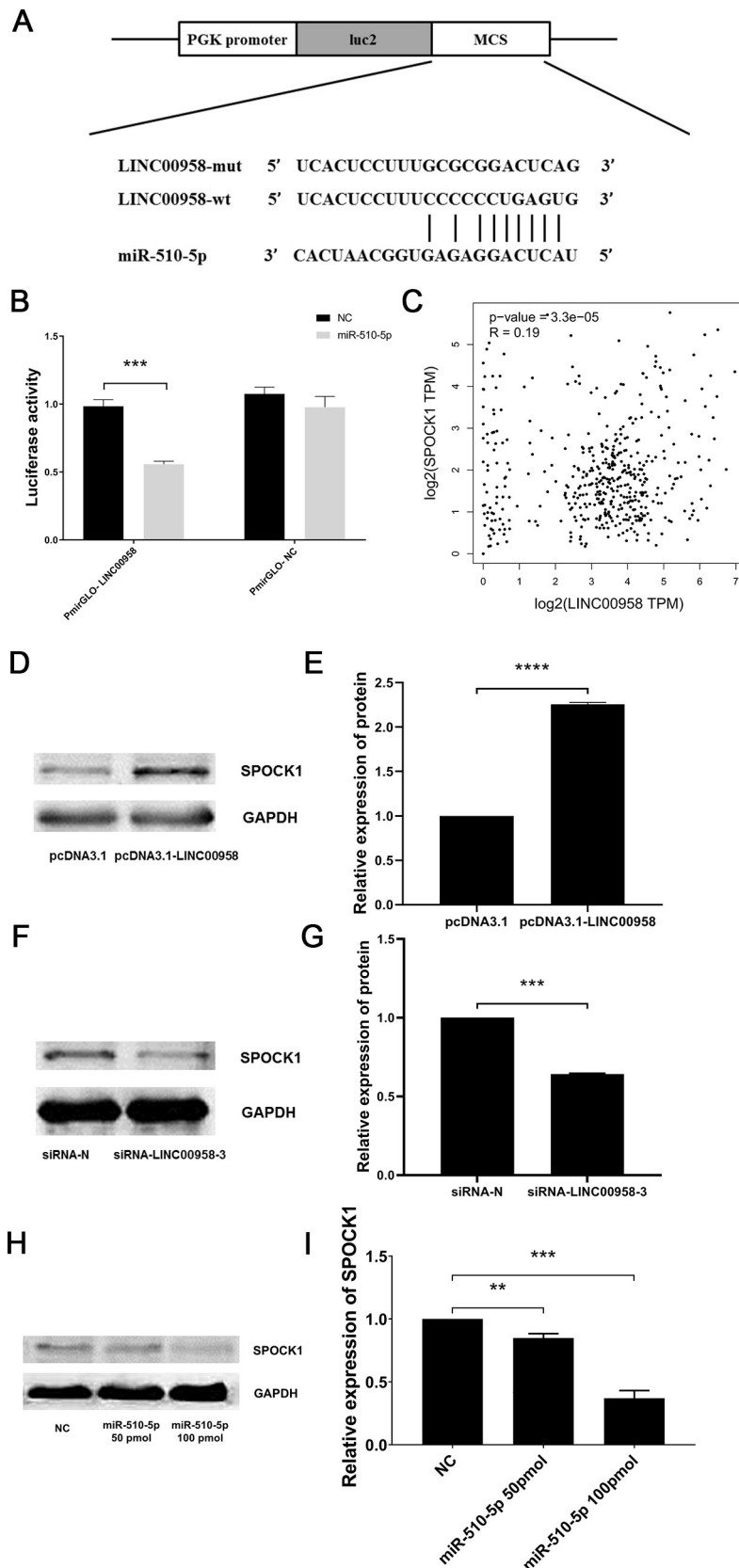

**Fig 4. *LINC00958* targeted miR-510-5p to regulate SPOCK1 expression.** (A) The targeting relationship between *LINC00958* and miR-510-5p was predicted by the ENCORI website. (B) A dual luciferase reporter gene experiment was used to verify the targeting relationship between *LINC00958* and miR-510-5p. (C) The GEPIA website predicted a correlation between *LINC00958* and SPOCK1. (D, E, F and G) The relationship between *LINC00958* and SPOCK1 was verified by a western blot. (H and I) The relationship between miR-510-5p and SPOCK1 was verified by a western blot. (** $P < 0.01$, *** $P < 0.001$, and **** $P < 0.0001$, n = 3).

functions in ESCC. For example, Xiaopeng et al. reported that SPOCK1 promotes the migration and invasion of EC109 cells through the EMT pathway [20]. Miao et al. reported that SPOCK1 affects the EMT pathway by acting on the transforming growth factor-β1 (TGF-β) [38]. Yusaku et al. reported that silencing of SPOCK1 by small interfering RNA inhibited ESCC cells migration and invasion [39] Therefore, we suggested that *LINC00958* might regulate the expression of SPOCK1 by its sponging of miR-510-5p, and involve proliferation, migration, invasion and EMT of ESCC. However, these results provided only a small glimpse into the complex functions of *LINC00958* in EC. Using the ENCORI website, we predicted that there could be as many as 52 possible miRNAs competing with *LINC00958*, and we only verified a small sample (S1 Table). In conjunction with the TargetScan website, we also predicted that there was a targeting relationship among a variety of *LINC00958*, miRNAs and mRNAs. For example, the adsorption of miR-185-5p, miR-625-5p/CPSF7 axis and miR-625-5p/LRRC8 axis have been confirmed, although the feasibility of the rest needs subsequent verification (S2 Table) [15,18,40].

## Conclusions

In this article, we investigated the role of *LINC00958* in ESCC and verified LINC00958 promoted proliferation, migration, invasion and EMT of ESCC via regulating miR-510-5p. It is hoped that our results will provide new strategies for the treatment of EC.

## Supporting information

**S1 Fig. CeRNA network of *LINC00958*, miRNA and mRNA interaction was predicted using the ENCORI and TargetScan websites.**
(TIF)

**S2 Fig. *LINC00958* is mainly concentrated in the cytoplasm of EC109 cells.**
(TIF)

**S3 Fig. CCK8 detected the effect of *LINC00958* on the proliferation of EC109 cells.**
(TIF)

**S1 Table. Interactions of *LINC00958* and miRNA, as predicted by the ENCORI website.**
(XLS)

**S2 Table. Interaction of *LINC00958*, miRNA and mRNA, as predicted by the TargetScan website.**
(XLSX)

**S1 Raw images.**
(PDF)

## Author Contributions

**Conceptualization:** Biqi Wang, Duo Tang, Zijia Liu, Qian Wang, Shan Xue, Zijie Zhao, Dongdong Feng, Chao Sheng, Jintao Li, Zhixiang Zhou.

**Data curation:** Biqi Wang, Duo Tang, Shan Xue, Zijie Zhao.

**Formal analysis:** Biqi Wang, Duo Tang, Dongdong Feng, Chao Sheng.

**Funding acquisition:** Zhixiang Zhou.

**Investigation:** Biqi Wang, Duo Tang, Zijia Liu, Qian Wang.

**Methodology:** Biqi Wang, Duo Tang, Jintao Li.

**Project administration:** Zhixiang Zhou.

**Resources:** Jintao Li, Zhixiang Zhou.

**Software:** Dongdong Feng, Chao Sheng.

**Validation:** Zijia Liu, Qian Wang, Shan Xue, Zijie Zhao.

**Visualization:** Zijia Liu, Qian Wang.

**Writing – original draft:** Biqi Wang, Duo Tang, Zijia Liu, Qian Wang, Dongdong Feng, Chao Sheng.

**Writing – review & editing:** Biqi Wang, Duo Tang, Shan Xue, Zijie Zhao, Jintao Li, Zhixiang Zhou.

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
