## [Decision Letter · Decision Letter 0]

26 Jan 2021

PONE-D-20-41067

LINC00958 upregulated SPOCK1 expression to promote the development of esophageal squamous cell carcinoma by sponging miR-510-5p

PLOS ONE

Dear Dr. Wang,

Thank you for submitting your manuscript to PLOS ONE. After careful consideration, we feel that it has merit but does not fully meet PLOS ONE’s publication criteria as it currently stands. Therefore, we invite you to submit a revised version of the manuscript that addresses the points raised during the review process.

Acceptance for publication is based on criteria which can be accessed on our website. These include

1. Conclusions must be supported and based on the data presented.

2. The manuscript must be written in standard English.

As this point, the manuscript contains statements that are not supported by experimental data including that LINC00958 contributes to ESCC development through upregulation of SPOCK, which would require to show that overexpression of LINC00958 indeed leads to the upregulation of SPOCK, but more so additional experimentation to assess the oncogenic effects. Please, see more detailed comments by the reviewers. 

The aspect of sponging is also not supported by the data presented. 

Additionally, quantification of Western Blot or new endpoints are needed for experiments where data show minor changes, but the effect is interpreted to be the causal to ESCC cell proliferation.

Extensive language editing is necessary.

Please, review our full criteria for publication on the PLOS ONE website.

Please submit your revised manuscript by April 1, 2021. If you will need more time than this to complete your revisions, please reply to this message or contact the journal office at plosone@plos.org. Please include the following items when submitting your revised manuscript:

We look forward to receiving your revised manuscript.

Kind regards,

Claudia D. Andl, Ph.D.

Academic Editor

PLOS ONE

Journal Requirements:

Reviewers' comments:

Reviewer's Responses to Questions

**Comments to the Author**

1. Is the manuscript technically sound, and do the data support the conclusions?

Reviewer #1: Partly

Reviewer #2: Partly

2. Has the statistical analysis been performed appropriately and rigorously? 

Reviewer #1: Yes

Reviewer #2: Yes

3. Have the authors made all data underlying the findings in their manuscript fully available?

Reviewer #1: Yes

Reviewer #2: Yes

4. Is the manuscript presented in an intelligible fashion and written in standard English?

Reviewer #1: Yes

Reviewer #2: Yes

5. Review Comments to the Author

Reviewer #1: In their manuscript “LINC00958 upregulated SPOCK1 expression to promote the development of esophageal squamous cell carcinoma by sponging miR-510-5p”, Wang, et al. determine that LINC00958 is upregulated in esophageal squamous cell carcinoma and evaluate the consequences of this phenomenon. Given the growing body of evidence implicating long noncoding RNAs in cancer development, this manuscript is of interest to the field of esophageal biology. However, I have significant concerns regarding the mechanism proposed in this work that must be addressed through major revision prior to publication of this article.

Major concern:

The data are too preliminary to support the central conclusion of this manuscript – that LINC00958-mediated ‘sponging’ of miR-510-5p results in elevated SPOCK1 expression, driving ESCC development. More work is needed to support this claim. The following questions need to be addressed to conclude that LINC00958 contributes to ESCC development through upregulation of SPOCK1:

1. Does SPOCK1loss-of-function reduce the oncogenic effects (growth, invasion, EMT) of LINC00958 overexpression?

2. Does SPOCK1 overexpression rescue the oncogenic effects (growth, invasion, EMT) of LINC00958 silencing?

3. Does LINC00958 overexpression result in an upregulation of SPOCK1?

The following questions need to be addressed in order to conclude that LINC00958 “sponging” of miR-510-5p is resulting in the SPOCK1-mediated oncogenic effects described in this manuscript:

1. Does overexpression of miR-510-5p reduce the oncogenic effects (growth, invasion, EMT) of LINC00958 overexpression?

2. Does miR-510-5p loss-of-function rescue the oncogenic effects (growth, invasion, EMT) of LINC00958 silencing?

Additionally, more evidence must be given to conclude that miR-510-5p and LINC00958 interact.

Minor concerns:

1. The manuscript requires significant editing for clarity and grammar.

2. Some of the data images are of poor quality.

3. Additional concerns that are articulated on a figure-by-figure basis below:

Figure 1:

The conclusion “These data indicated that LINC00958 may play an important role in the occurrence and development of esophageal cancer” is not supported by the data. These data only indicate that LINC00958 is upregulated in ESCC compared with normal esophagus.

Figure 2:

EC109 cells already have elevated LINC00958 expression, which dampens the impact of panels E, G, and I. Demonstrating that LINC00958 expression increases the growth rate of the normal SHEE cell line that does not already have elevated LINC00958 expression would be more convincing.

C: Most of the differences in the cell index between the two samples occur within the first ten hours, which is too short of a time period for these differences to be a result of increased proliferation. After the first ten hours, the two samples exhibit the same rate of proliferation.

C, D: Both siRNA-N and PCDNA3.1-N should exhibit similar growth rates given that both are EC109 cells transfected with negative controls. The discrepancy between the two growth rates should be addressed.

E, F: The colony formation assays are overgrown, preventing the accurate quantification of single colonies. A better-quality image is needed.

Figure 3:

Similar to the comment for figure 2: These experiments should be performed in SHEE cells that overexpress LINC00958.

A: What do the scale bars represent?

E: Need a higher quality western blot.

Figure 4:

More discussion is required to explain why miR-510-5p and SPOCK1 were evaluated in this manuscript.

D: Need a higher quality western blot.

E: SPOCK1 is misspelled in the Y-axis label.

Reviewer #2: Esophageal cancer is a highly lethal malignancy with a 5-year overall survival rate less than 20%. In this manuscript, Wang et al. described the roles of a long non-coding RNA, LINC00958, in esophageal cancer. LINC01296 was shown to promote esophageal squamous cell carcinoma cell proliferation and invasion previously (Am J Cancer Res. 2018; 8(10): 2020–2029). Using a combination of in vitro experiments, the authors showed a potential oncogenic role of LINC00958 via competing with miR-510-5p to regulate SPOCK1 expression. My major concern is that there is no in vivo experiments in this study. I also would recommend the authors to request a native English speaker to reedit this manuscript.

1. In figure 1A, there are several horizontal lines, do they represent medium value or mean of the TCGA data? It needs to describe clearly in the figure legend.

2. In figure 2G-I, what is the mechanism of inducing G1 cell cycle arrest? Are there any apoptosis-dependent cell death here?

3. The authors demonstrated a little effect on the ESCC cell proliferation by overexpression or knockdown of LINC00958 (Figure 2C-D) within 3 days. This minimal effect most likely will go away in the long term if you look at the trend of the curve. In vivo experiments need to be performed to show if manipulation of LINC00958 has any meaningful effect.

4. In the figure 3, the effect of LINC00958 on EMT proteins is also minimal based on the WB results, suggesting LINC00958 may not be the major regulator of ESCC invasion.

6. PLOS authors have the option to publish the peer review history of their article (what does this mean?). If published, this will include your full peer review and any attached files.

Reviewer #1: No

Reviewer #2: No

---

## [Author Response · Author response to Decision Letter 0]

1 Apr 2021

Dear reviewers and editor,

Re: Resubmission of manuscript reference No.PONE-D-20-41067.

Thank you for inviting us to submit a revised draft of our manuscript entitled, " LINC00958 upregulated SPOCK1 expression to promote the development of esophageal squamous cell carcinoma by sponging miR-510-5p" to PLOS ONE. We also appreciate the time and effort you and each of the reviewers have dedicated to providing insightful feedback on ways to strengthen our paper. Thus, it is with great pleasure that we resubmit our article for further consideration. We have incorporated changes that reflect the detailed suggestions you have graciously provided. Since it has been reedited by native English speaking editors, the title is now changed to “LINC00958 upregulates SPOCK1 expression to promote the development of oesophageal squamous cell carcinoma by sponging miR-510-5p”. We also hope that our edits and the responses we provide below satisfactorily address all the issues and concerns you and the reviewers have noted. If there is anything inappropriate in the revision, please point it out and we will revise it seriously. 

To facilitate your review of our revisions, the following is a point-by-point response to the questions and comments delivered in your letter dated Jan 26 2021.

Editor’s Suggestions:

1. [Conclusions must be supported and based on the data presented. The manuscript contains statements that are not supported by experimental data including that LINC00958 contributes to ESCC development through upregulation of SPOCK, which would require to show that overexpression of LINC00958 indeed leads to the upregulation of SPOCK, but more so additional experimentation to assess the oncogenic effects.]

RESPONSE: Thank you for providing these insights. We agree with you and have incorporated this suggestion throughout our paper. We also improved the expression of the experimental data in the subsequent revision of the manuscript. Western blotting assay was added to determine the effect of LINC00958 on the expression of SPOCK1.The results showed that the over-expression of LINC00958 increased the expression of SPOCK1, while the Knock-down expression of LINC00958 decreased the expression of SPOCK1. The data showed that LINC00958 could regulate the expression of SPOCK1. (page 14, lines 277-281)

2. [The aspect of sponging is also not supported by the data presented.]

RESPONSE: The high expression of LINC00958 has an important relationship with the occurrence and development of esophageal cancer. In order to verify the function ways of LINC00958 in ESCC, we predicted up to 52 miRNAs competing with LINC00958 in ENCORI website (S1 Table), but we only verified the interested part. Then, we have predicted the competitive binding relationship between LINC00958 and miR-510-5p through the ENCORI website (Fig 4A). A dual luciferase reporter gene experiment was used to verify that the miR-510-5p mimics significantly inhibited the luciferase activity of the PmirGLO-LINC00958 plasmid, but had no effect on the PmirGLO-NC plasmid (Fig 4B). The results showed that LINC00958 sponging miR-510-5p. Chen et al. ( Downregulated microRNA-510-5p acts as a tumor suppressor in renal cell carcinoma), and Sun et al. (CircRNA SCARB1 Promotes Renal Cell Carcinoma Progression Via Mir- 510-5p/SDC3 Axis) have reported that miR-510-5p acts as a tumor suppressor in renal cell carcinoma, inhibiting cell proliferation and migration, and promoting cell apoptosis. According to our experimental results, LINC00958 acts through the miR-510-5p/SPOCK1 axis. It can be speculated that miR-510-5p may act as a tumor suppressor in ESCC and play a similar role to that in renal cell carcinoma. Through the Target Scan website, we predicted the targeting relationship between miR-510-5p and SPOCK1, and the experimental results also showed that miR-510-5p was the upstream miRNA of SPOCK1. Therefore, we proposed that LINC00958 could regulate the expression of SPOCK1 by adsorbing miR-510-5p through sponge to promote the development of ESCC. However, these results are just the tip of the iceberg for the complex function of LINC00958 in cancer. Combined with the Target Scan website, we also predicted the targeting relationships among various LINC00958, miRNAs and mRNAs, such as the absorption of miR-185-5p, miR-625-5p/ CpsF7 and miR-625-5p/ LRRCC8 axis, which have been confirmed, and the feasibility of the rest still needs subsequent verification (S2 Table).

3. [ Quantification of Western Blot or new endpoints are needed for experiments where data show minor changes, but the effect is interpreted to be the causal to ESCC cell proliferation.]

RESPONSE: Thank you for your suggestions. We conducted new experiments on Western Blot and new antibodies were used to ensure higher quality images and to make the data more descriptive. Our results indicated that LINC00958 could regulate the migration and invasion by the EMT pathway (Fig 3E, F, G and H) and the expression of SPOCK1 (Fig 4D, E, F and G) in ESCC.

4. [ Extensive language editing is necessary. The manuscript must be written in standard English.]

RESPONSE: Thank you for your suggestion. In order to improve the manuscript, we have edited for proper English language, grammar, punctuation, spelling, and overall style by one or more of the highly qualified native English speaking editors at International Research Promotion English Language Editing Services (IRP-ELES). The Certificate Verification Key is IRP-2021-ELES-26672.

5. [If applicable, we recommend that you deposit your laboratory protocols in protocols.io to enhance the reproducibility of your results.]

RESPONSE: Thank you for your advice. We will deposit our laboratory protocols in protocols.io according to the standards.

6. [Please ensure that your manuscript meets PLOS ONE's style requirements, including those for file naming.]

RESPONSE: Thank you for your suggestions. We have modified the name of the manuscript and the document according to the style requirements of PLOS ONE. The title is now changed to “LINC00958 upregulates SPOCK1 expression to promote the development of oesophageal squamous cell carcinoma by sponging miR-510-5p”.

7. [PLOS ONE now requires that authors provide the original uncropped and unadjusted images underlying all blot or gel results reported in a submission’s figures or Supporting Information files.]

RESPONSE: Thank you for your advice. We have submitted the original uncropped and unadjusted blot and gel results images as the supporting information (S1_raw_images).

Reviewer#1 Suggestions:

[Major concern]

1. [Does SPOCK1loss-of-function reduce the oncogenic effects (growth, invasion, EMT) of LINC00958 overexpression?]

RESPONSE: Thank you for your affirmation of our work, and thank you very much for your valuable suggestions. You raise an important question. The western blot results showed that the expression of SPOCK1 significantly increased when LINC00958 was overexpressed (Fig 4D and 4E), and significantly decreased when LINC00958 was downregulated （Fig 4F and 4G). The function of SPOCK1 has been thoroughly studied. Yusaku et al. (Regulation of SPOCK1 by dual strands of pre-miR-150 inhibit cancer cell migration and invasion in the growth of squamous cell carcinoma) and Song et al. (Up-regulation of SPOCK1 induces epithelial-mesenchymal transition and promotes migration and invasion in esophageal squamous cell carcinoma) have reported that SPOCK1 can regulate the migration and invasion of esophageal squamous cell carcinoma through EMT, while the overexpression or functional loss of SPOCK1 will affect the migration and invasion of esophageal squamous cell carcinoma.

2. [Does SPOCK1 overexpression rescue the oncogenic effects (growth, invasion, EMT) of LINC00958 silencing?]

RESPONSE: Thanks for your advice. Changes in the expression of LINC00958 will lead to changes in the expression of SPOCK1, which has been verified by western blot experiment (Fig 4D and E). SPOCK1 could regulate the migration and invasion of esophageal squamous cell carcinoma through EMT, while the loss of function of SPOCK1 would affect the migration and invasion of esophageal squamous cell carcinoma. We also confirmed the upstream and downstream relationship between LINC00958 and SPCOK1 through dual luciferase assay, suggesting that LINC00958 can indeed regulate the function of esophageal squamous cell carcinoma by acting on SPOCK1. LINC00958 is a long and complex lncRNA, and we have predicted hundreds of possible functions. LINC00958-SPOCK1 is one of the ones we have verified successfully, but this segment may also be the tip of the iceberg to reveal the function of LINC00958. For example, LINC-ROR is a well-studied lncRNA in esophageal squamous cell carcinoma, which can promote the progression of esophageal squamous cell carcinoma through MDM2, SOX9 and other pathways. Thank you for your criticism and correction. We will improve the results of the paper. Please continue to pay attention to us.

3. [Does LINC00958 overexpression result in an upregulation of SPOCK1?]

RESPONSE: Thanks for your suggestion. We did the relevant Western blot experiment, which proved that when LINC00958 was overexpressed, the expression of SPOCK1 was significantly up-regulated, and when LINC00958 was knocked down, the expression of SPOCK1 was significantly down-regulated. (page14, lines277-281)

4. [Does overexpression of miR-510-5p reduce the oncogenic effects (growth, invasion, EMT) of LINC00958 overexpression?]

RESPONSE: Thank you for your suggestion. According to our experimental results, overexpression of LINC00958 will promote the growth, migration, invasion and EMT of esophageal squamous cell carcinoma cells, while knocking down LINC00958 will get the opposite result (Page 12, lines 243-259). The dual luciferase experiment was used to verify that miR-510-5p mimics significantly inhibited the luciferase activity of PmirGLO-LINC00958, but had no effect on PmirGLO-NC (Fig 4B) indicated that there was a competitive action relationship between LINC00958 and miR-510-5p. Therefore, the overexpression of miR-510-5p can reduce the oncogenic effect of LINC00958 through the competitive binding of LINC00958.

5. [Does miR-510-5p loss-of-function rescue the oncogenic effects (growth, invasion, EMT) of LINC00958 silencing?]

RESPONSE: Thanks for your suggestion, Choi et al. (MicroRNA library screening identifies growth-suppressive microRNAs that regulate genes involved in cell cycle progression and apoptosis) have been identified that miR-510-5p has tumor suppressor function and can inhibit cell growth. The dual luciferase assay proved that LINC00958 had a competitive effect with miR-510-5p (page13, lines271-274). Theoretically, when the function of miR-510-5p was lost, it could no longer combine with LINC00958 competitively. Thus, the oncogenic effects of LINC00958 silencing can be rescue.

6. [More evidence must be given to conclude that miR-510-5p and LINC00958 interact.]

RESPONSE: Thanks for your suggestion. We have predicted that there are as many as 52 possible miRNAs competing with LINC00958 in ENCORI, and we only verify the parts of our interest (S1 Table). We paid attention to miR-510-5p that had an active relationship with LINC00958 (Figure 4A). After that, the dual luciferase experiment was used to verify that miR-510-5p mimics significantly inhibited the luciferase activity of PmirGLO-LINC00958, but had no effect on PmirGLO-NC (Fig 4B), indicating that miR-510-5p and LINC00958 indeed have an interaction relationship.

[Minor concerns]

1. [The manuscript requires significant editing for clarity and grammar.]

RESPONSE: Thank you for your suggestion. In order to improve the manuscript, we have edited for proper English language, grammar, punctuation, spelling, and overall style by one or more of the highly qualified native English speaking editors at International Research Promotion English Language Editing Services (IRP-ELES). The Certificate Verification Key is IRP-2021-ELES-26672.

2. [Some of the data images are of poor quality.]

RESPONSE: Thank you for your suggestions. We have made improvements for the pictures with poor quality in the article, and we hope they can meet the standards. 

[Figure 1]

[The conclusion “These data indicated that LINC00958 may play an important role in the occurrence and development of esophageal cancer” is not supported by the data. These data only indicate that LINC00958 is upregulated in ESCC compared with normal esophagus.]

RESPONSE: Thank you for your suggestion. LINC00958 is highly expressed in clinical samples and routine cell models of esophageal squamous cell carcinoma in the laboratory, and it is indeed impossible to confirm its role in the occurrence and development. We will change this result to indicate that LINC00958 is upregulated in ESCC, which may be related to the development of EC (page 10, lines 205-208).

[Figure 2]

[EC109 cells already have elevated LINC00958 expression, which dampens the impact of panels E, G, and I. Demonstrating that LINC00958 expression increases the growth rate of the normal SHEE cell line that does not already have elevated LINC00958 expression would be more convincing.]

RESPONSE: Thank you for your advice. EC109 or other esophageal squamous cell carcinoma cells, such as EC9706 and KYSE150, all showed high expression of LINC00958 (Fig 1B). The function of LINC00958 was verified by overexpression and knockdown. In EC109 cells, the overexpression of LINC00958 under the condition of high expression did not significantly lower the knockdown effect, but the purpose of this was to make the overexpression and knockdown confirm each other and improve the reliability of data and conclusions. SHEE cells were not selected because they were esophageal immortalized cells, which had the phenotype of normal esophageal cells and could not be used to characterize esophageal cancer. 

[C: Most of the differences in the cell index between the two samples occur within the first ten hours, which is too short of a time period for these differences to be a result of increased proliferation. After the first ten hours, the two samples exhibit the same rate of proliferation.]

RESPONSE: Thank you for your suggestion. We used RTCA for proliferation determination. Real-time cell analyzing (RTCA) systems use gold microelectrode biosensors in each well of microtiter plates to measure electrical impedance. The electrical potential creates an electrical field between the cells and microelectrodes. Increasing the number of adherent cells and changing conditions in the cell culture alter the impedance. The impedance gives quantitative information about the number, viability, morphology and migration of the cells. The impedance of gold microelectrodes in RTCA systems when cells are not present or not adhered onto the electrodes is determined with ionic cell culture medium solution. Adherent cells act as an insulator on the surface of the electrode and change the ionic medium of the electrode solution, increasing the impedance ( Türker Şener L, Albeniz G, Dinç B, Albeniz I. iCELLigence real-time cell analysis system for examining the cytotoxicity of drugs to cancer cell lines. Exp Ther Med. 2017;14(3):1866-1870. doi:10.3892/etm.2017.4781). It detects cell proliferation by using mechanical components at the bottom of a petri dish (https://www.agilent.com/en/technology/cellular-impedance). It shows a lot of sensitivity when the cells are small, and the EC109 cells are in good condition, and when we cultured them, they were able to double in 12 hours, so this may have resulted in the same rate of proliferation 10 hours later. At the beginning, PCDNA3.1-LINC00958 group showed significant difference in proliferation compared with the control group PCDNA3.1-N, and we also conducted CCK8 experiment to prove this, so as to improve the reliability of the data (S3 Fig). 

[C, D: Both siRNA-N and PCDNA3.1-N should exhibit similar growth rates given that both are EC109 cells transfected with negative controls. The discrepancy between the two growth rates should be addressed.]

RESPONSE: Thank you for your advice. In C and D, we conducted several experiments and found that siRNA-N would indeed grow at the same rate as normal cells. However, the growth rate of PCDNA3.1-N was significantly lower than that of normal cells, because the total length of PCDNA3.1-N was 5428bp, which might increase the burden of cells after transplanting, thus reducing the growth rate.

[E, F: The colony formation assays are overgrown, preventing the accurate quantification of single colonies. A better-quality image is needed.]

RESPONSE: Thank you for your advice. We have carried out the experiment again and submitted a picture with better quality (Fig 2E and F).

[Figure 3]

[Similar to the comment for figure 2: These experiments should be performed in SHEE cells that overexpress LINC00958.]

RESPONSE: Thanks for your advice. EC109 or other esophageal squamous cell cells, such as EC9706 and KYSE150, all have a high expression of LINC00958. The reason why SHEE cell was not selected is that it is an Oesophageal squamous cell, which has the phenotype of normal esophageal cells, so it cannot be used to characterize esophageal cancer.

[A: What do the scale bars represent?]

RESPONSE: Thank you for your criticism and correction. The scale lines in figure 3A are intended to give the reader a more intuitive view of the boundaries of the scratches.

[E: Need a higher quality western blot.]

RESPONSE: Thank you for your suggestion. As for the Western Blot, we have used better antibodies to carry out the experiment again, so as to obtain higher quality pictures (Fig 3E).

[Figure 4]

[More discussion is required to explain why miR-510-5p and SPOCK1 were evaluated in this manuscript.]

RESPONSE: The high expression of LINC00958 has an important relationship with the occurrence and development of esophageal cancer. In order to verify the function ways of LINC00958 in ESCC, we predicted up to 52 miRNAs competing with LINC00958 in ENCORI website (S1 Table), but we only verified the interested part. We focus on miR-510-5p, and have predicted the competitive binding relationship between LINC00958 and miR-510-5p through the ENCORI website (Fig 4A).We used TargetScan to predict the downstream mRNA of miR-510-5p, and we focused on SPOCK1. In addition, it was predicted that there was a significant correlation between LINC00958 and SPOCK1 in esophageal cancer through GEPIA (Fig 4C). The western blot results showed that the expression of SPOCK1 significantly increased when LINC00958 was overexpressed (Fig 4D and 4E), and significantly decreased when LINC00958 was downregulated（Fig 4F and 4G). We synthesized miR-NC and miR-510-5p mimics and transfected them into EC109 cell. Compared with the control group, miR-510-5p significantly inhibited the expression of SPOCK1 (Fig 4H and 4I), so miR-510-5p and SPOCK1 were evaluated in our manuscript. In addition, we will conduct more specific experiments to explain the related functions of miR-510-5p and SPOCK1. Please keep following us.

[D: Need a higher quality western blot.]

RESPONSE: Thanks for your advice. We have carried out the experiment again and obtained higher quality pictures (Fig 4H).

[E: SPOCK1 is misspelled in the Y-axis label.]

RESPONSE: Thanks for your advice and we have corrected the incorrect spelling (Fig 4I).

Reviewer#2 Suggestions:

[major concern]

[There is no in vivo experiments in this study. I also would recommend the authors to request a native English speaker to reedit this manuscript.]

RESPONSE: Thank you very much for your affirmation of our work, at the same time for your suggestions also express heartfelt thanks. As for the in vivo experiment, we have started to prepare for it and will publish it in the next article. Thank you for your suggestions and hope you can continue to pay attention to us. At the same time, In order to improve the manuscript, we have edited for proper English language, grammar, punctuation, spelling, and overall style by one or more of the highly qualified native English speaking editors at International Research Promotion English Language Editing Services (IRP-ELES). The Certificate Verification Key is IRP-2021-ELES-26672. We hope that the quality of our articles can meet the standard. 

1. [In figure 1A, there are several horizontal lines, do they represent medium value or mean of the TCGA data? It needs to describe clearly in the figure legend.]

RESPONSE: Thank you for your criticism and correction. The TCGA data represents the mean value, and we will indicate it in the picture. (page 11, line 210)

2. [In figure 2G-I, what is the mechanism of inducing G1 cell cycle arrest? Are there any apoptosis-dependent cell death here?]

RESPONSE: Thank you for your suggestion. The mechanism of LINC00958 inducing G1 block has not yet been identified as its downstream target. Through bioinformatics, we have predicted the downstream targets of LINC00958 have CDK5, CDK15, CDK18 and other genes regulating cell cycle (S1 Fig), but whether they are reliable still needs experimental verification. Please continue to pay attention to us. 

3. [The authors demonstrated a little effect on the ESCC cell proliferation by overexpression or knockdown of LINC00958 (Figure 2C-D) within 3 days. This minimal effect most likely will go away in the long term if you look at the trend of the curve. In vivo experiments need to be performed to show if manipulation of LINC00958 has any meaningful effect.]

RESPONSE: Thank you for your suggestion. We used RTCA for proliferation determination，Real-time cell analyzing (RTCA) systems use gold microelectrode biosensors in each well of microtiter plates to measure electrical impedance. The electrical potential creates an electrical field between the cells and microelectrodes. Increasing the number of adherent cells and changing conditions in the cell culture alter the impedance. The impedance gives quantitative information about the number, viability, morphology and migration of the cells. The impedance of gold microelectrodes in RTCA systems when cells are not present or not adhered onto the electrodes is determined with ionic cell culture medium solution. Adherent cells act as an insulator on the surface of the electrode and change the ionic medium of the electrode solution, increasing the impedance.( Türker Şener L, Albeniz G, Dinç B, Albeniz I. iCELLigence real-time cell analysis system for examining the cytotoxicity of drugs to cancer cell lines. Exp Ther Med. 2017;14(3):1866-1870. doi:10.3892/etm.2017.4781) It is through a dish at the bottom of the original mechanical test cell proliferation (https://www.agilent.com/en/technology/cellular-impedance). It shows a lot of sensitivity when the cells are small, and the EC109 cells are in good condition, and when we cultured them, they were able to double in 12 hours, so this may have resulted in the same rate of proliferation 10 hours later. At the beginning, PCDNA3.1-LINC00958 group showed significant difference in proliferation compared with the control group PCDNA3.1-N, and we also conducted CCK8 experiment to prove this, so as to improve the reliability of the data (S3 Fig). We are preparing for the in vivo experiment, which will be reported in the next article. Please continue to pay attention to us.

4. [In the figure 3, the effect of LINC00958 on EMT proteins is also minimal based on the WB results, suggesting LINC00958 may not be the major regulator of ESCC invasion.]

RESPONSE: Thank you for your suggestion. We use Odyssey's fluorescent color development method, which leads to the background of imprinting is not as clear as that of chemical (HRP) color development method, thus causing great interference to statistics. We have carried out the experiment again and submitted the picture with better quality (Fig 3E and 3G). It is clear indicated that the downregulation of LINC00958 could increase the expression of E-Cadherin, Occludin and ZO-1 and could decrease the expression of Vimentin (Fig 3E and 3F). The upregulation of LINC00958 could reduce the expression of E-Cadherin, Occludin and ZO-1, and increase the expression of Vimentin (Fig 3G and 3H). These results all indicated that LINC00958 could regulate the migration and invasion by the EMT pathway in ESCC. Thank you for your correction.

---

## [Decision Letter · Decision Letter 1]

27 Apr 2021

PONE-D-20-41067R1

LINC00958 upregulates SPOCK1 expression to promote the development of oesophageal squamous cell carcinoma by sponging miR-510-5p

PLOS ONE

Dear Dr. Wang,

Thank you for submitting your manuscript to PLOS ONE. After careful consideration, we feel that it has merit but does not fully meet PLOS ONE’s publication criteria as it currently stands. Therefore, we invite you to submit a revised version of the manuscript that addresses the points raised during the review process.

Overall, the revised version is improved and the manuscript strengthened, which both reviewers agreed upon. However, point 4 of the PLOS ONE publication criteria require that conclusions are supported by the data, yet the conclusion that LINC00958 is promoting growth by regulating SPOCK1 expression via its sponging of miR-510-5p is still not well supported by the data at this point. Ideally, more experimentation would be recommended to strengthen these claims prior to publication. Instead, I would suggest to focus on modifications in the presentation of the data and conclusions.

For acceptance, it would be sufficient to change the title to focus on the role of LINC00958 instead of SPOCK1 and revising the statements mentioned by reviewer 1 which are not supported by the present data. 

We look forward to receiving your revised manuscript.

Kind regards,

Claudia D. Andl, Ph.D.

Academic Editor

PLOS ONE

Journal Requirements:

Additional Editor Comments (if provided):

Overall, the reviewers agreed that the revised version is improved and most concerns were addressed by the authors. However, point 4 of the PLOS ONE publication criteria require that conclusions are supported by the data and this point the authors’ conclusion that LINC00958 is promoting growth by regulating SPOCK1 expression via its sponging of miR-510-5p is still not well supported by the data. More experimentation would be required to strengthen these claims prior to publication: The newly added data demonstrate that LINC00958 is regulating SPOCK1 expression, but the magnitude is small and further evidence is drawn from the literature and published changes in SPOCK1 in regard to ESCC proliferation and EMT. The authors would be required to demonstrate that SPOCK1 knockdown or overexpression rescue the phenotypes observed as result of LINC00958 knockdown or overexpression.

I would like to suggest to instead change the title to focus on the role of LINC00958 in directly measured outcomes, and focus on suggestions from reviewer 1 to:

Include a legend for the graph in panel A.

Change the writing to not overstate the results and drawn conclusion, e.g., LINC00958 regulates migration and the EMT pathway. However, more work is needed to conclude that the observed decrease in migration following LINC00958 modulation is specifically by the EMT pathway. Remove the statements regarding the role of SPOCK1 in ESCC to reflect that the experiments do not address this question.

Reviewers' comments:

Reviewer's Responses to Questions

**Comments to the Author**

1. If the authors have adequately addressed your comments raised in a previous round of review and you feel that this manuscript is now acceptable for publication, you may indicate that here to bypass the “Comments to the Author” section, enter your conflict of interest statement in the “Confidential to Editor” section, and submit your "Accept" recommendation.

Reviewer #1: (No Response)

Reviewer #2: (No Response)

2. Is the manuscript technically sound, and do the data support the conclusions?

Reviewer #1: Partly

Reviewer #2: (No Response)

3. Has the statistical analysis been performed appropriately and rigorously? 

Reviewer #1: Yes

Reviewer #2: (No Response)

4. Have the authors made all data underlying the findings in their manuscript fully available?

Reviewer #1: Yes

Reviewer #2: (No Response)

5. Is the manuscript presented in an intelligible fashion and written in standard English?

Reviewer #1: Yes

Reviewer #2: (No Response)

6. Review Comments to the Author

Reviewer #1: Wang, et al provide compelling evidence that LINC00958 is upregulated in ESCC and promotes ESCC proliferation and cell migration. Given the growing interest in the tumor promoting role for long noncoding RNAs in ESCC, this manuscript is of interest to the field. Further, the authors have addressed many of the concerns raised during the initial review and have produced a stronger manuscript. However, the authors’ conclusion that LINC00958 is promoting growth by regulating SPOCK1 expression via its sponging of miR-510-5p is still not well supported by the data. More experimentation is required to strengthen these claims prior to publication.

Major concern:

1. One of the central conclusions of the manuscript – that LINC00958 is contributing to ESCC cell proliferation and migration by suppressing SPOCK1 expression via ‘sponging’ of miR-510-p – is not well supported by the data. While the authors have now added data that demonstrate that LINC00958 is regulating SPOCK1 expression, these data are not particularly convincing given the small (~25%) magnitude of the changes. While the authors argue that changes in SPOCK1 have been well-studied in ESCC and are associated with proliferation and EMT, this evidence likely does not extend to the small changes in SPOCK1 expression observed during LINC00958 over or underexpression. The authors need to address this discrepancy, ideally by determining if SPOCK1 knockdown or overexpression rescue the phenotypes observed by LINC00958 knockdown or overexpression.

Minor concerns:

1. Panel A – Please include a legend for this graph.

2. Some conclusions need to be rewritten to be supported by the data:

“These results all indicated that LINC00958 could regulate the migration and invasion by the EMT pathway” (line 257-259).

These results indicate that LINC00958 regulates migration and the EMT pathway. However, more work is needed to conclude that the observed decrease in migration following LINC00958 modulation is specifically by the EMT pathway. Please discuss.

“Thus, it was clear that SPOCK1 had a cancer-promoting function in ESCC” (285-286).

These experiments do not address the role of SPOCK1 in ESCC. Please remove this conclusion.

Reviewer #2: (No Response)

7. PLOS authors have the option to publish the peer review history of their article (what does this mean?). If published, this will include your full peer review and any attached files.

Reviewer #1: No

Reviewer #2: No

---

## [Author Response · Author response to Decision Letter 1]

1 May 2021

Dear Dr. Andl and reviewers,

Re: Resubmission of manuscript reference PONE-D-20-41067R1.

Thank you for inviting us to submit a revised version of the manuscript entitled, " LINC00958 upregulates SPOCK1 expression to promote the development of oesophageal squamous cell carcinoma by sponging miR-510-5p" to PLOS ONE. Thank you for your approval of our previous revision, we also appreciate the time and effort you have dedicated to providing insightful feedback on ways to strengthen our paper. Thus, it is with great pleasure that we resubmit our article for further consideration. We have incorporated changes that reflect the detailed suggestions you have graciously provided. The title has now been changed to “LINC00958 promotes proliferation, migration, invasion, and Epithelial-Mesenchymal Transition of oesophageal squamous cell carcinoma cells”. We also hope that our revision and the responses we provide below satisfactorily address all the issues and concerns you and the reviewers have noted. If there is anything inappropriate in the revision, please point it out and we will revise it seriously.

Thank you for your consideration. We look forward to hearing from you.

Kind Regards,

Biqi Wang, MA.Eng, +86 18811071427

Faculty of environment and life, Beijing University of Technology, Beijing 100124, China

Email: 1585835414@qq.com

To facilitate your review of our revisions, the following is a point-by-point response to the questions and comments delivered in your letter dated April 27, 2021.

Journal Requirements:

1. [Please review your reference list to ensure that it is complete and correct. If you have cited papers that have been retracted, please include the rationale for doing so in the manuscript text, or remove these references and replace them with relevant current references. Any changes to the reference list should be mentioned in the rebuttal letter that accompanies your revised manuscript. If you need to cite a retracted article, indicate the article’s retracted status in the References list and also include a citation and full reference for the retraction notice.]

RESPONSE: We have reviewed the reference list and no retracted papers were cited in the manuscript. The changes to the reference list are as follows:

1. We’ve rescinded the use of the first quotation, and replace it with the current citation 1. Sung H, Ferlay J, Siegel RL, Laversanne M, Soerjomataram I, Jemal A, et al. Global cancer statistics 2020: GLOBOCAN estimates of incidence and mortality worldwide for 36 cancers in 185 countries. CA Cancer J Clin. 2021.countries. due to the update of data. (page 2, line 41)

2. We’ve removed the previous citation [23] and change it to citation [39] which is cited in the discussion. (page 16, line 340)

Additional Editor Comments:

1. [However, point 4 of the PLOS ONE publication criteria require that conclusions are supported by the data and this point the authors’ conclusion that LINC00958 is promoting growth by regulating SPOCK1 expression via its sponging of miR-510-5p is still not well supported by the data. More experimentation would be required to strengthen these claims prior to publication: The newly added data demonstrate that LINC00958 is regulating SPOCK1 expression, but the magnitude is small and further evidence is drawn from the literature and published changes in SPOCK1 in regard to ESCC proliferation and EMT. The authors would be required to demonstrate that SPOCK1 knockdown or overexpression rescue the phenotypes observed as result of LINC00958 knockdown or overexpression. I would like to suggest to instead change the title to focus on the role of LINC00958 in directly measured outcomes.]

RESPONSE: Thank you for your suggestions. We agree with you and have changed the title of the article to “LINC00958 promotes proliferation, migration, invasion, and Epithelial-Mesenchymal Transition of oesophageal squamous cell carcinoma cells”. (page 1, lines 4-6)

As for the question of the small magnitude of LIN00958 regulating the expression of SPOCK1, it have intuitively shows that the overexpression of LINC00958 significantly increases the expression of SPOCK1, while the knockdown of LINC00958 significantly reduces the expression of SPOCK1 in Fig. 4D and 4F. We used the ImageJ software again to conduct quantitative analysis on Fig. 4D 4F, and the results as shown in Fig. 4E and 4G were obtained.

2. [Focus on suggestions from reviewer 1 to:Include a legend for the graph in panel A.]

ROSPONSE:Thank you for your suggestion.We have added the legend for the graph in panel A. (Fig 1A) (pages 10-11, lines 210-212)

3. [Change the writing to not overstate the results and drawn conclusion, e.g., LINC00958 regulates migration and the EMT pathway. However, more work is needed to conclude that the observed decrease in migration following LINC00958 modulation is specifically by the EMT pathway. Remove the statements regarding the role of SPOCK1 in ESCC to reflect that the experiments do not address this question.]

RESPONSE: Thank you for your advice. We have changed the writing in the manuscript and have removed the statement regarding the role of SPOCK1 in ESCC. The specific modifications are as follows:

1. The results indicated there was a high expression of LINC00958 in ESCC, which promoted proliferation, migration, invasion and Epithelial–Mesenchymal Transition (EMT) process of ESCC cells, and this effect may be via regulating miR-510-5p. (page 2, lines 35-37)

2. These results indicated that LINC00958 could regulate migration, invasion and EMT of ESCC.(page 13, lines 256-257)

3. Therefore, we suggested that LINC00958 might regulate the expression of SPOCK1 by its sponging of miR-510-5p, and involve proliferation, migration, invasion and EMT process of ESCC. (page 16, lines 340-342)

Reviewer#1 Suggestions:

[Major concern]

1. [One of the central conclusions of the manuscript – that LINC00958 is contributing to ESCC cell proliferation and migration by suppressing SPOCK1 expression via ‘sponging’ of miR-510-p – is not well supported by the data.]

RESPONSE: Thank you for your suggestion. In our manuscript, it can be concluded that LINC00958 can regulate the proliferation and migration of oesophageal squamous cell cancer cells, and LINC00958 can regulate the expression of SPOCK1 by sponging miR-510-5p, but the conclusion that LINC00958 is contributing to ESCC cell proliferation and migration by suppressing SPOCK1 expression via ‘sponging’ of miR-510-p is indeed not sufficiently supported by the data. After our discussion, the title of the manuscript has been changed to " LINC00958 promotes proliferation, migration, invasion, and Epithelial-Mesenchymal Transition of oesophageal squamous cell carcinoma cells".

2. [While the authors have now added data that demonstrate that LINC00958 is regulating SPOCK1 expression, these data are not particularly convincing given the small (~25%) magnitude of the changes. While the authors argue that changes in SPOCK1 have been well-studied in ESCC and are associated with proliferation and EMT, this evidence likely does not extend to the small changes in SPOCK1 expression observed during LINC00958 over or underexpression. The authors need to address this discrepancy, ideally by determining if SPOCK1 knockdown or overexpression rescue the phenotypes observed by LINC00958 knockdown or overexpression.] 

RESPONSE: Thank you for your advice. The western blot results have intuitively shows that the expression of SPOCK1 significantly increased when LINC00958 was overexpressed (Fig 4D), and significantly decreased when LINC00958 was downregulated (Fig 4F). We used the ImageJ software again to conduct quantitative analysis on Fig. 4D and 4F, and get a significant magnitude of change as shown in Fig. 4E (~125%) and 4G (~36%) were obtained. At the same time, we have changed the statement of the conclusion to: “These results indicated that LINC00958 could regulate migration, invasion and EMT of ESCC.” (page 13, lines 256-257) “The results indicated that LINC00958 could regulate the expression of SPOCK1 by competing with miR-510-5p in ESCC.” (page 14, lines 282-283) “Therefore, we suggested that LINC00958 might regulate the expression of SPOCK1 by its sponging of miR-510-5p, and involve proliferation, migration, invasion and EMT process of ESCC.” (page 16,lines 340-342)

[Minor concerns:]

1. [Panel A – Please include a legend for this graph.]

RESPONSE: Thank you for your suggestion.We have added the legend for the graph in panel A. (Fig 1A) (pages 10-11, lines 210-212)

2. [Some conclusions need to be rewritten to be supported by the data:“These results all indicated that LINC00958 could regulate the migration and invasion by the EMT pathway” (line 257-259). These results indicate that LINC00958 regulates migration and the EMT pathway. However, more work is needed to conclude that the observed decrease in migration following LINC00958 modulation is specifically by the EMT pathway. Please discuss.]

RESPONSE: Thank you for your suggestion. We have rewritten the conclusion to “These results indicated that LINC00958 could regulate migration, invasion and EMT of ESCC.”(page 13, lines 256-257) “We also detected an upregulated expression of LINC00958 in ESCC cells, and verified that LINC00958 promoted proliferation, migration, invasion and EMT process of ESCC.” (page 15,lines 318-320)

3. [“Thus, it was clear that SPOCK1 had a cancer-promoting function in ESCC” (285-286).These experiments do not address the role of SPOCK1 in ESCC. Please remove this conclusion.]

RESPONSE: Thank you for your suggestion. We have removed the conclusion about SPOCK1 had a cancer promoting funtion in ESCC, and have rewritten the conclusion to “Therefore, we suggested that LINC00958 might regulate the expression of SPOCK1 by its sponging of miR-510-5p, and involve proliferation, migration, invasion and EMT process of ESCC.” (page 16, lines 340-342)

---

## [Editor Report · Decision Letter 2]

4 May 2021

LINC00958 promotes proliferation, migration, invasion, and Epithelial-Mesenchymal Transition of oesophageal squamous cell carcinoma cells

PONE-D-20-41067R2

Dear Dr. Wang,

We’re pleased to inform you that your manuscript has been judged scientifically suitable for publication and will be formally accepted for publication once it meets all outstanding technical requirements.

Kind regards,

Claudia D. Andl, Ph.D.

Academic Editor

PLOS ONE
---

## [Editor Report · Acceptance letter]

10 May 2021

PONE-D-20-41067R2 

*LINC00958* promotes proliferation, migration, invasion, and Epithelial-Mesenchymal Transition of oesophageal squamous cell carcinoma cells 

Dear Dr. Wang:

I'm pleased to inform you that your manuscript has been deemed suitable for publication in PLOS ONE. Congratulations! Your manuscript is now with our production department. 

Kind regards, 

on behalf of

Dr. Claudia D. Andl 

Academic Editor

PLOS ONE